# Internal cohesion gradient as a novel mechanism of collective cell migration

**Larissa M. Oprysk, Maribel Vazquez** **\*, Troy Shinbrot**

Department of Biomedical Engineering Rutgers, The State University of New Jersey, Piscataway, New Jersey, United States of America

\* mv582@soe.rutgers.edu

## Abstract

Experiments demonstrate that individual cells that wander stochastically can migrate persistently as a cluster. We show by simulating cells and their interactions that collective migration by omnidirectional cells is a generic phenomenon that can be expected to arise whenever (a) leading and trailing cells migrate randomly, and (b) leading cells are more closely packed than trailing neighbors. The first condition implies that noise is essential to cluster motion, while the second implies that an internal cohesion gradient can drive external motion of a cluster. Unlike other swarming phenomena, we find that this effect is driven by cohesion asymmetry near the leading cell, and motion of interior cells contribute minimally – and in fact interfere with – a cluster's persistent migration.

## Author summary

Cell replacement therapy in the nervous system is an emerging therapeutic strategy to restore lost functionality due to injuries or disease. Recent research has shown promise for many conditions including spinal cord injury, neurological diseases, and retinal degeneration. The retina provides an excellent model for studying neural transplantation due to its immuno-privilege environment and the direct connection to the brain. In this therapy, Stem or progenitor cells, capable of differentiating into any neural type, are inserted into the host tissue where they must then migrate toward specific areas of damage and integrate within the existing neural network. However, studies have illustrated mixed success due in large part to lack of appropriate migration by the replacement cells within columnar networks formed by retinal neurons. Experimental studies of neural progenitor migration have shown that groups of cells moving collectively demonstrate highly directional movement toward external concentration gradients compared to individual cells. In this paper, we computationally explore a novel mechanism of collective migration to demonstrate persistent directional movement toward the existing neural network for integration. We show that by connecting individual cells in a cluster and varying the strength of their attachments to one another, the cluster tends to migrate in the direction of the strongest cohesive attachment within the cluster. Understanding these key mechanisms of cellular migration will provide the ability to control replacement cell migration and ultimately lead to more effective transplantation outcomes.

**Data availability statement:** All MATLAB code is available within the supplemental materials.

**Funding:** This work was supported by the National Science Foundation (CBET 2243644 to MV). The funders had no role in study design, data collection and analysis, decision to publish, or preparation of the manuscript.

**Competing interests:** The authors have declared that no competing interests exist.

## Introduction

Collective cell migration is essential to development and metabolism of organisms extending from the most ancient bacteria and molds to the most advanced mammals [1]. At the oldest end of this range, millions of E. coli cooperate to produce intricate swarming behaviors [2], and Dictyostelia self-organize into differentiated fruiting bodies [3]. More complex organisms rely on collective migration for everything from gastrulation [4] and tubulation [5] during embryogenesis to wound healing [6] in fully developed tissues. On the pathological side, collective cell migration is intrinsic to the formation of tumors [7] and fistulas [8], as well as other dysfunctional tissue remodeling associated with disease and inflammation [9].

Despite its prevalence, the dynamics of collective cell migration are striking in their complexity – for example generating pulsatile growth fronts in tumors [10] and oscillatory ruffles on healing membranes [11]. Understanding the underlying mechanisms is also important for practical applications, for example informing tissue regeneration therapies. Research in cell replacement therapy shows promise for a variety of neurological diseases, spinal cord injuries, and retinal degeneration. In this therapy, stem or progenitor cells, which can differentiate into the desired neuronal cell type, are inserted into a host tissue and must then migrate toward areas of damage, effectively integrate within the native environment, and form novel synaptic connections to recreate the neural pathway, restoring the damaged tissue [12–14]. However, preclinical studies have illustrated mixed success due in large part to the lack of donor cell migration toward and integration within the native network [15,16]. The underlying mechanisms that control donor cell migration and infiltration within adult tissue environments are incompletely understood.

We have previously shown *in vitro* that in the presence of a fibroblast growth factor (FGF) signaling gradient, individual retinal progenitor cells (RPCs) migrate seemingly randomly, while clusters of cells under the same conditions exhibit persistent motion toward the gradient maximum [17,18]. Typical data, shown in Fig 1, reveal that single cells in the presence of an

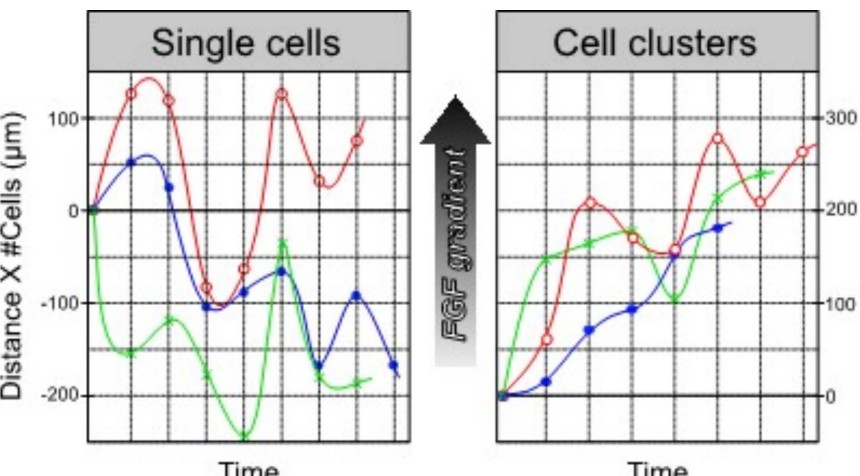

**Fig 1.** *Experimental migration trajectories toward higher FGF concentration (indicated by the black arrow) for single cells (left) and clusters of  ~10 Drosophila retinal progenitor cells in microchannels prepared with nearly steady FGF gradient (upward in plot orientation). A chemotactic index (CI) greater than 0.5 indicates positive directional migration along concentration gradients [17,19]. Small clusters of cells tend to migrate toward the higher FGF concerta-tion (Average CI = 0.61 ± 0.15) while individual cells appear to migrate more randomly (Average CI = 0.22 ± 0.20). Each plot shows 3 trials (raw data provided in Pena 2019 supplemental information [17]); distances traveled are normalized to the number of cells in the cluster. The curves are cubic splines to aid the eye.*

FGF gradient wander without apparent direction, while cell clusters under the same conditions migrate persistently toward a gradient maximum (upward in the Fig). Similar results have been reported in mouse RPCs [19,20].

Intriguingly, the growth factors examined in both fly and mouse RPCs upregulate the expression of cadherins [19–21]: trans-membrane proteins that mediate cell-cell cohesion. These growth factors do not appear to affect persistent migration of individual cells; hence we investigate here the mechanism by which internal, cell-cell, interactions affect net motion of a cluster.

This role is of both clinical and fundamental relevance. Clinically speaking, the goal of intentionally directing progenitors to desired targets (e.g., to regions of a damaged retina), relies on establishing what factors affect migration and how the factors function. While many studies have demonstrated cell-induced chemotaxis in individual cells [22–24], our studies indicate that individual RPCs do not chemotax without influence from neighboring cells [17,19,25]. Moreover, the data was measured in ultra-low concentration gradients that suppress cell ability to create their own local gradient by consuming/degrading the surrounding chemoattractant field. The underlying mechanisms by which a net external force is produced by interactions between individual cells – that do not individually exhibit directional migration – challenge our understanding [26].

Some headway into resolving the paradox that interactions between cells connected within a cluster can generate an additional external force (arising from cell-cell connections) that individual cells do not feel can be made by observing that a gradient in internal cell-cell cohesion breaks the symmetry between what we'll term the "front" (higher FGF) and "back" (lower FGF) of a collection of cells. This is informative since Noether's theorem [26,27] dictates that symmetries (here of cohesive potential energies) are always associated with conservation laws (here of forward momentum), and so our task is to analyze how asymmetries of internal forces can generate a net external force.

For both fundamental and practical reasons, then, we examine conditions under which internal, inter-cellular, interactions – mediated here by membrane cadherins – can produce net motion of collections of cells. Before beginning, we note that remarkable as this may seem, multiple organisms, including protozoans [28] and skinks (sand-swimming lizards) [29] are known to accomplish a related paradoxical effect, termed metaboly. In these and other examples [30–32] simulations have shown that metaboly uses internal deformations to generate deterministic migration. As we'll show, however, collective motion can also be achieved using non-deterministic and rectilinear motion of cluster components.

We utilize Agent-Based Modeling (ABM), a first-principles approach that defines cells as autonomous "agents" that follow a set of "rules" (e.g., Newton's laws of motion) based on known mechanisms or observed behaviors, reviewed in [33]. ABM approaches have identified emergent behavior [34,35] in the collective motion of individual cells and bacteria as well as plant and animal development and morphogenesis [36], and a small number of studies have described ABM in the context of stem cell migration [37,38].

In our ABM simulation, we analyze the dynamics of a cluster of cells acted on by differential cohesion along the length of the cluster in response to an external concentration gradient (e.g., of FGF). We emphasize that substrate stresses are assumed to be identical along the length of the cluster: the net force toward increased gradient emerges only from asymmetry of internal cohesive forces combined with stochastic wandering. This is potentially important for future applications of retinal transplantation, as the environment associated with degenerative disorders of the eye can result in dysregulation of cell-matrix interactions [39]. As we will describe, this internal asymmetry leads to net external motion of a cluster through an unexpected and subtle mechanism.

## Results and discussion

### Model description

We use a one-dimensional agent-based model to explore the mechanical effects of differential cohesion strength across a cluster of cells on the cluster's ability to collectively migrate. The decision to use a 1D model is motivated by growth factor gradients developing in one direction and enables direct comparison between *in-silico* and *in-vitro* models. We represent individual cells as spheres arranged in a line constrained to a planar surface, where each sphere is mechanically linked to its nearest neighbors by damped springs in one of two configurations: a) Kelvin-Voigt and b) Standard Linear Solid (SLS) configuration. Both of these models are commonly used to describe viscoelastic cells [40,41]. The simpler Kelvin-Voigt model is represented by a purely elastic spring and a purely viscous damper in parallel, shown schematically in Fig 2A, and obeying:

$$\sigma = K\varepsilon + \eta\dot{\varepsilon}. \tag{1}$$

Here $\sigma$ and $\varepsilon$ denote stress and strain respectively, $\dot{\varepsilon}$ is the rate of strain, K is an elastic modulus and $\eta$ is a viscosity. Cellular elasticity is further broken down into tensile and compressive stresses [42]. Tensile stress is generated by intercellular connections, mediated by adherens including cadherins [43], and intracellular cytoskeletal networks [44,45] that depend on cell type. Compressive stresses are generated by hydrostatic and cytoskeletal responses to deformation [46].

We distinguish between tensile and compressive stresses in our model by setting $K = K_t + K_c$, where $K_t$ defines the tensile modulus, and $K_c$ defines the compressive

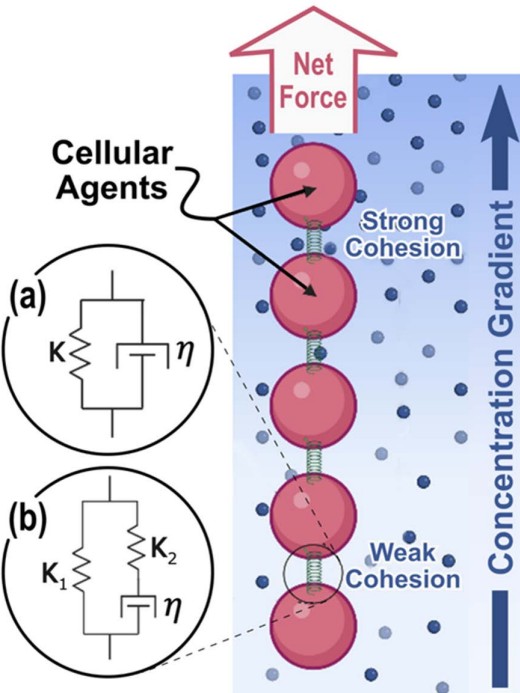

**Fig 2. Schematic of 1D Agent Based Model (ABM) for 5 cell cluster.** *Concentration gradient influences intercellular cohesion to produce a net force on cluster. Similar results are obtained using either* **(a)** *Kelvin-Voigt or* **(b)** *Standard Linear Solid models of intercellular interactions.*

modulus. Algorithmically, $K_t = 0$ when cells are compressed, and $K_c = 0$ when cells are separated.

The viscous term similarly is determined by two effects – first, irreversible structural changes within each cell, and second viscous interaction with the cellular environment [47]. Correspondingly we write $\eta\dot\varepsilon = \eta_i\dot\varepsilon_i + \eta_e\dot\varepsilon_e$, where $\eta_i\dot\varepsilon_i$ is a viscous stress that depends on relative velocities, $\dot\varepsilon_i$, of adjacent cells and $\eta_e\dot\varepsilon_e$ depends on velocity of a cell in the lab frame. Thus, cells moving relative to one another feel a viscosity $\eta_i$, determined by their cytoplasm, while cells moving in a surrounding fluid feel a viscosity $\eta_e$ determined by the fluid.

The Standard Linear Solid representation is slightly more involved and includes a creep term that Kelvin-Voigt lacks. Creep is included using a second spring in series with the viscous damper, as shown schematically in Fig 2B, so that the spring with modulus $K_1$ always resists deformation, while the other spring relaxes with characteristic time $\tau = \eta / K_2$ as the damper creeps.

This model can be written in the so-called "Maxwell" form as:

$$\sigma + \tau\,\dot\sigma = K_1\varepsilon + \tau\left(K_1 + K_2\right)\dot\varepsilon, \tag{2}$$

where as before the elastic moduli are different under tension and compression, and the viscous time, $\tau$, involves both inter-cellular and extra-cellular (environmental) terms. To numerically integrate Eq. (2), we rewrite each term as derivatives of a common displacement, $x_{(ij)}$, representing the distance between sequential, $i$ th and $j$ th, cells:

$$\dddot{x}_{(ij)} = \frac{K_{1(ij)}}{\tau_{(ij)}}x_{(ij)} + \left(K_{1(ij)} + K_{2(ij)}\right)\dot{x}_{(ij)} - \frac{1}{\tau_{(ij)}}\ddot{x}_{(ij)}. \tag{3}$$

This is then numerically solved using the 4$^{th}$ order Runge Kutta integration, and (assuming equal mass cells) a force given by the sum $\ddot{x}_{(i-1,i)} + \ddot{x}_{(i,i+1)}$ is applied to each cell from both forward and rearward neighbors. Although Eq. (3) is third order, we find that the combined internal and external damping effectively limits numerical instabilities, additional details provided in the Methods section.

In our model, both the strengths of repulsion and viscosity between adjacent agents are constant for all cells in a cluster. Tensile components of $K_1$ and $K_2$, however, are assumed to increase linearly along the length of the cluster. Biologically speaking, the local growth factor concentration is what promotes the upregulation of cadherin expression within individual cells, and the cells with higher levels of cadherin expression enable stronger attachments [48,49]. Based on this, we argue the gradient of spring stiffness would be directly correlated to the gradient of growth factor concentration in the environment (linear growth factor gradient leads to linear cohesion gradient). This assumption is based on experimental data mentioned earlier (cf. Fig 1) demonstrating that clusters of retinal progenitor cells exhibit cadherin-mediated chemotaxis in the presence of a FGF gradient [17,19]. This increase in tensile strength is intended to model a response to a gradient in an external factor. Individual cells in the higher growth factor concentration environment experience more upregulation of cell-cell adhesion proteins which lead to more stable and stronger adherens junction stability [48,49]. This differential cell-cell attraction strength will be referred to as the internal cohesion gradient herein.

We emphasize that although each cellular agent exhibits a different tensile modulus toward the front and the back of the cluster, it is not evident that this difference in *internal* modulus alone will produce a net *external* force, affecting transport of the cluster. Indeed, as we will show through detailed simulations, net chemotaxis only occurs under a limited

set of conditions. Interestingly, we find first that one of these conditions is that the cells must migrate stochastically. It is known that most cell types migrate in either using adhesion -dependent (mesenchymal) [50] or -independent (amoeboid) mechanisms [51,52]. However, if migration is deterministically activated (i.e., cells move in alternating inward and outward directions periodically in time), no collective motion of clusters is seen in our simulations. On the other hand, if migration of individual cells is invoked stochastically (i.e., cells moving in either direction randomly), persistent forward motion of clusters emerges.

## Deterministic Model: Differential cohesion alone does not produce cluster motion

Baseline simulations were conducted to evaluate the effect of differential cohesion on collective migration under strictly deterministic conditions – i.e., without stochastic variations. The expectation is that cohesion is an internal force to a cluster and should not affect net external migration. We find that this is so in the simplest case, but by analyzing this case in detail, we find clues to subtle behaviors that can produce net migration.

We begin with the case shown in Fig 3A, where N = 5, representative of a small cluster as in experiments of Fig 1, simulated cells are initialized in a linear configuration from an initially relaxed state with no initial velocity and linearly increasing cohesion strength from bottom ($N_1$) to top ($N_5$), where the top cells' cohesion are 10 times stronger than the bottom cells (additional key parameters are given in the caption Fig 3). Periodically, the two outermost cells ($N_1$ and $N_5$) are shifted an equal distance away from the center of mass and given time to relax, where cells return to a zero-stress state (which typically occurs when they are just touching their neighbors), under the influence of cohesive interaction with their nearest neighbor (see methods). As expected, the cells with the strongest tensile moduli (near the leading edge of the cluster) deviate the least from the center of mass while the cells with the weakest tensile moduli (near the trailing edge) deviate the most from the center of mass. The center of mass of the cluster (dashed line), however, remains stationary. The same outcome is found for all parameter values, displacement periods and amplitudes. Additionally, we confirm that the center of mass remains stationary if the outermost cells are moved toward and away from the center of mass, either in phase (Fig 3B) or antiphase (Fig 3C).

## Analysis: deterministic case vs. stochasticity on the outermost cells

In detail, what occurs within the cluster is sketched in Fig 4 for the simplest case of a 3-cell cluster. Starting from an equally spaced initial state ("Initial" in Fig 4), if we extend the outer cells ("Extension") and then allow the cells to relax ("Relaxation"), we see that the upper cells, with stronger cohesion, relax rapidly, while the lower cell relaxes more slowly. For illustrative purposes, we choose the extension distance to be two cell radii in the Fig.

As indicated to the right of the Fig, in the limit of very strong cohesion above and very weak cohesion below, the two upper cells rapidly relax to their average position, 1 radius from the initial state, and the one lower cell remains nearly at its extended position, displaced by 2 radii. More generally, if the central body itself consists of $n$ cells, each of unit mass, then $n+1$ cells will be displaced upward by $2/(n+1)$ radii, and it is easily confirmed that the center of mass is consequently unchanged. The simple calculation is based on the expectation, sketched in Fig 4 and confirmed in Fig 3, that higher cohesion cells in a cluster remain nearby, while a lower cohesion neighbor strays further away. If the motions are deterministic – i.e., if cells are periodically displaced by fixed distances – the system adopts an equilibrium between upward and downward forces, and the center of mass remains stationary.

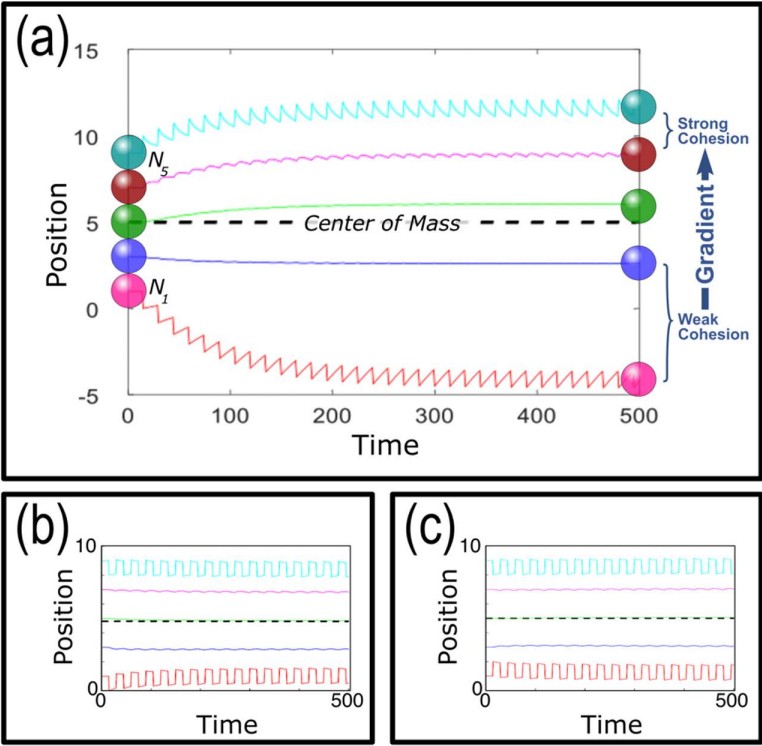

**Fig 3. Simulations of 5-cell clusters with intercellular cohesion gradient.** *(a) The outermost cells, $N_1$ & $N_5$, experience an outward extension of the same distance at the same time, and each colored line shows subsequent positions. The periodic extension causes the cluster to elongate until it reaches a natural equilibrium state. The black dashed line shows the center of mass, which remains stationary. (b) The same simulation in which the outermost cells move both up and down in phase, and (c) the same in which the outermost cells move in and out in phase. As expected, in all cases the center of mass remains stationary except for small transient oscillations. Key parameters are as follows: gradient strength (ratio of strongest to weakest cohesion strengths) = 10, cluster size = 5, protrusion amplitude (length of protrusions) = 1, protrusion period (frequency of protrusions) = 15. Protrusion amplitude is a fixed value in this simulation, whereas all other simulations have a variable amplitude using a Gaussian whose standard deviation is the specified protrusion amplitude.*

Remarkably, the situation changes if cells extend stochastically. To understand why, let's allow the leading and the trailing cells (upper and lower as shown in Fig 4) to take *random* steps forward or back. For the time being, we only allow the outermost cells to take steps. This allows direct comparison with Fig 3 and agrees with recent findings that leader cell specification emerges from the existence of an unconstrained free edge which enables extended protrusion formation. This initiates a positive molecular feedback loop that promotes further protrusion formations [53]. We will allow all cells to migrate after analyzing this simpler case.

For now, we analyze cluster motion in the limit that leading cells relax very rapidly and trailing cells relax very slowly. As depicted in Fig 5A, the leading cell in a cluster takes steps forward or back at stochastic distances defined by a Gaussian with standard deviation $\sigma$ indicated by dashes in the plots to the right of the cells. Our analysis is purely computational, but physiologically speaking this could represent the effect of pseudopod or adhesion formation [54–56].

If protrusions are short compared to the distance to the nearest cell, inward (extending the cell inward to the center of the cluster) and outward (extending the cell away from the cluster center) protrusions will be equally likely. Arrows in Fig 5 indicate mean distances, $\pm\sqrt{2/\pi}\ \sigma$, which have equal magnitude toward and away from the cluster. On average, this produces the same equilibrium state as depicted in Fig 4.

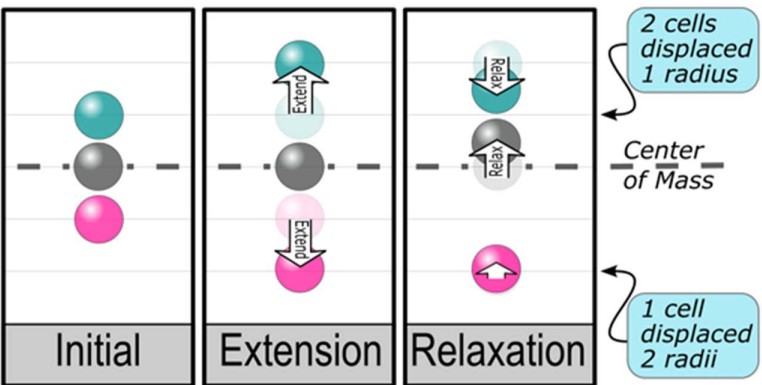

**Fig 4. *Analysis of motion from Fig 3A due to single outward step of outermost cells, starting form an "Initial" state at rest.*** *During "Extension" phase, the upper and lower cells are instantaneously extended outward (here by 2 cell radii), leaving the central cell body (gray) and the center of mass (dashed) stationary. During the subsequent "Relaxation" phase, the lower cell, feeling very weak cohesive tension with its neighbor, moves and affects the central cell body little, while the upper cell, subject to strong tension, rapidly equilibrates to an average position with the neighboring central cell body. After each cycle of extension and relaxation, the bottom cell is displaced 2 radii, and the two upper cells are displaced 1 radius, leaving the center of mass unmoved.*

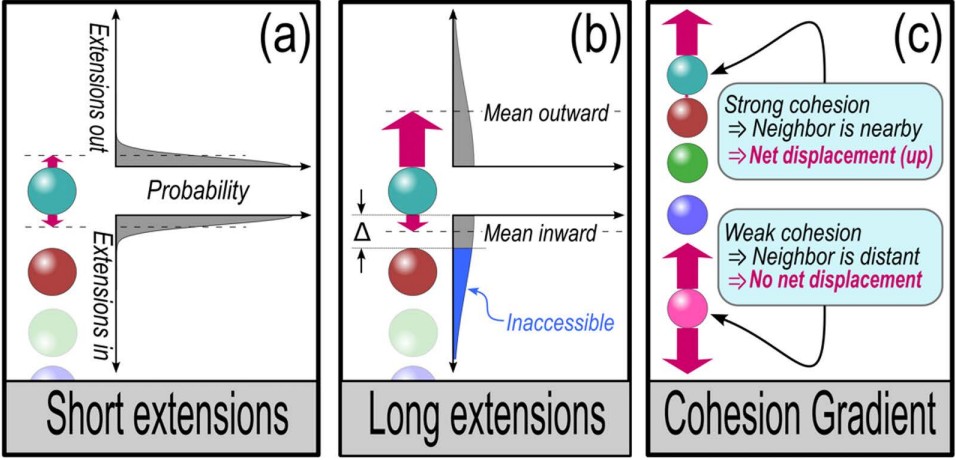

**Fig 5. *Stochastic extension of outermost cells – (a) For stochastic extension distances (red arrows) that are short compared with the distance to the nearest cell, inward and outward motions are equally probable.*** *Dashed lines indicate mean extension distances, $\pm\sqrt{2/\pi}\,\sigma$ , where σ is the standard deviation of a Gaussian governing extension probabilities, shown to the right. (b) Extensions cannot reach beyond a neighboring cell a distance Δ away, so as extension distances grow (or distances between neighbors shrink), extension probabilities will be truncated, as shown in blue. Consequently, the mean inward distance traveled by the outermost cell will become shorter than the mean outward distance. (c) In the presence of a cohesion gradient, the stronger cohesion end of a cluster will exhibit closer neighboring cells, and so will lead asymmetric migration of the cluster. The weaker cohesion end will exhibit more distant neighbors, and so will generate more symmetric, smaller displacement, migration.*

The situation changes, however, if protrusions become long compared with the distance to the nearest neighbor. Assuming that pseudopods cannot extend past a neighboring cell, the furthest a protrusion – and hence its defining Gaussian – can extend is a distance Δ to the neighbor, indicated in Fig 5(b). We label extensions beyond the neighbor "inaccessible," and correspondingly the mean distance that extensions will reach inward (up in the plot shown) will be shortened compared with the distance outward. This means that for longer

protrusions, a cell at the boundary of a cluster typically extends further away from, rather than toward, the cluster, as indicated by asymmetric red arrows panel (b).

In a cluster with a cohesion gradient, sketched in Fig 5(c), cells will tend to be more closely spaced toward the higher cohesion end, and so the differential protrusion lengths described in panel (b) are more pronounced at that end. As indicated by red arrows in the sketch, it is easy to construct a scenario in which the low cohesion end of a cluster feels symmetric inward and outward forces, while the high cohesion end is consistently pulled away from the cluster.

## Quantification: mathematical derivation of cluster mean displacement

It is straightforward to calculate protrusion lengths assuming a normal half-Gaussian distribution, $f(x,\sigma)$, of protrusions with variance $\sigma^2$: $f(x,\sigma) = \dfrac{\sqrt{2}}{\sigma\sqrt{\pi}}\exp\left(-\dfrac{x^2}{2\sigma^2}\right)$ for $x \geq 0$.

The mean displacement of a cell inward, excluding the inaccessible region produced by a neighbor a distance $\Delta$ from the edge of a cell as indicated in Fig 5(b), is:

$$Mean-to-neighbor\,|_\Delta = \frac{\int_0^\Delta x f(x,\sigma)\,dx}{\int_0^\Delta f(x,\sigma)\,dx} = \sqrt{\frac{2}{\pi}}\sigma\,\frac{1-\exp\left(-\dfrac{\Delta^2}{2\sigma^2}\right)}{\mathrm{Erf}\left(\dfrac{\Delta}{\sqrt{2}\sigma}\right)}, \tag{4}$$

so the net protrusion distance of a cell with one side free and one side constrained at distance $\Delta$ is

$$\langle disp\rangle_\Delta = Mean-to-neighbor\,|_\infty - Mean-to-neighbor\,|_\Delta = \sqrt{\frac{2}{\pi}}\sigma\left[1+\frac{\exp\left(-\dfrac{\Delta^2}{2\sigma^2}\right)-1}{\mathrm{Erf}\left(\dfrac{\Delta}{\sqrt{2}\sigma}\right)}\right]. \tag{5}$$

By virtue of the leading (trailing) cell being strongly (weakly) tethered to its neighbors, each time it extends a protrusion, cohesive tension from its neighbors will rapidly (slowly) draw it back, as we saw before in the "Relaxation" stage of Fig 4. In the limit of an extremely weakly tethered trailing cell, it will on average travel nearly independently from the rest of the cells in a cluster, and so we set $\Delta \gg \sigma$, and Eq. (5) becomes $\langle disp\rangle_{\Delta\gg\sigma} \to 0$. For a strong cohesion gradient, on the other hand, the leading cell will attain $\langle disp\rangle_{\Delta\ll\sigma} \to \sqrt{2/\pi}\,\sigma$, and will asymmetrically pull the cluster.

This means that in the limit of a very strong cohesion gradient, we expect a cluster of mass $N$ to be displaced on average once per protrusion time of the leading cell, so that the cluster displacement will approach:

$$\langle disp\rangle \to \frac{\langle disp\rangle_{\Delta\ll\sigma}}{N} = \frac{\sqrt{1+(n+1)\sqrt{\dfrac{1}{n+1}}}}{n+2}\sqrt{D_o t} = \sqrt{\frac{2}{\pi}}\,\frac{\sigma}{N}\left[1+\frac{\exp\left(-\dfrac{\Delta^2}{2\sigma^2}\right)-1}{\mathrm{Erf}\left(\dfrac{\Delta}{\sqrt{2}\sigma}\right)}\right]. \tag{6}$$

The expected behavior defined by Eq. (6) applies in the limit that leading and trailing cohesion are respectively very strong and very weak. Actual displacements can be slower than this, and this is an average, which will apply over long times or, assuming ergodicity, over many clusters.

## Qualitative confirmation: cluster migration with and without an internal gradient

We test Eq. (6) by replacing the constant displacements of the outermost cells shown in Fig 3 with zero mean Gaussian displacements. We will shortly examine the effect of stochastic motion of all cells in a cluster, but we start with the simpler case shown already in Fig 3, where only the outer cells extend. Biologically speaking, this is motivated by the expectation that inner cells will be crowded and so cannot extend, and so migrate, far from their neighbors: this turns out to be an important consideration in cluster migration. Since a Gaussian could in principle displace a cell past its nearest neighbor, we cut off the resulting extension whenever the inward displacement contacts that neighbor. The leading and trailing cells thus obey zero mean Gaussian that are truncated whenever the cell would contact a neighbor. All agents other than the outermost ones follow the rules already defined, and so respond to stresses imparted by their neighbors.

In Fig 6A, we show a typical simulation (parameters in Fig caption) of a 5-cell cluster, exhibiting a mean upward migration of the center of mass toward the stronger cohesion end of the cluster. The Fig also shows persistent upward motion of the top 4 cells, above a nearly untethered trailing cell that, because of its low cohesion, wanders in a random walk almost independently of its neighbors.

We note that two effects are at work in Fig 6. First, the center of mass rises nearly steadily, and we will shortly compare its motion with Eq. (6). Second, the leading cell almost invariably extends forward (up), and then relaxes back (down), not the reverse. This qualitatively confirms the expectation that cells cannot migrate past their nearest neighbor, and so excursions by the leading cell tend to be outward. We recall that the leading cell is always closely

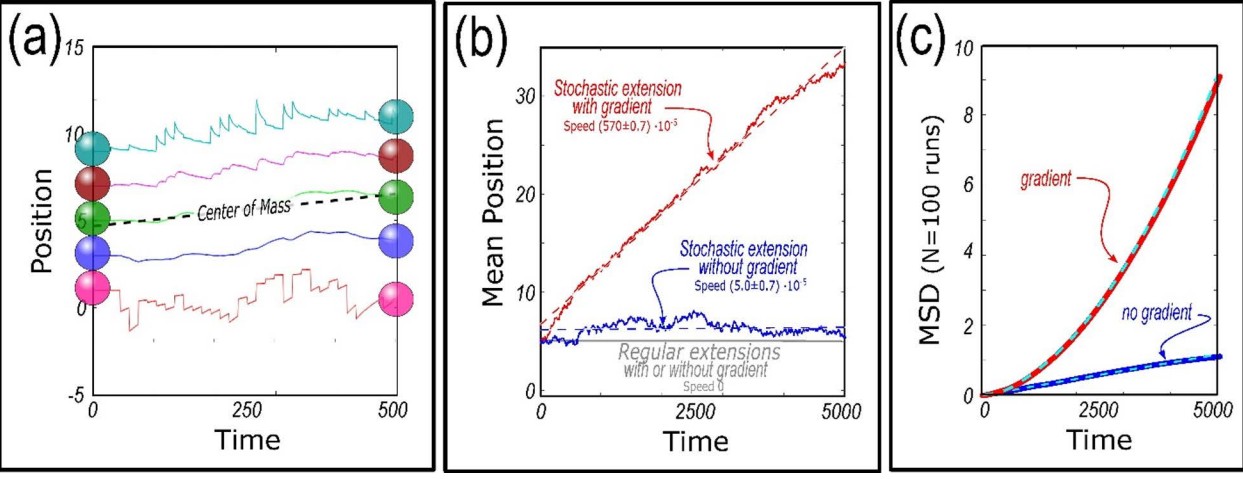

**Fig 6. Simulations of 5-cell cluster with Gaussian fluctuations.** *(a) Outermost cells, $N_1$ and $N_5$ experience displacements according to Gaussian fluctuations. The colored lines show the subsequent positions of each cell within the cluster. Note the upper 4 cells migrate consistently upward, toward the higher internal cohesion gradient, while the lower cell wanders almost independently. (b) Comparison between centers of mass subject to regular extensions (gray, from Fig 3), Gaussian fluctuations without a cohesion gradient (blue), and Gaussian fluctuations with a gradient (red). Solid lines indicate the average cluster position at successive times, and the dotted line are the least squares fit to the data. Only the case with the gradient migrates persistently, as shown by the positive slope (red dotted line). Key parameters are as follows: gradient strength = 10, cluster size = 5, protrusion amplitude = 1, protrusion period = 5. (c) Mean Squared Displacement analysis of N = 100 simulations of cell clusters without a gradient (blue) and with a gradient (red). Final positions range from -1 to 7 radii away from the initial position for clusters with a gradient vs ±3 radii away for clusters without a gradient. Cyan dashed lines indicate linear regression: $y = (231 \pm 0.2) \times 10^{-6} * x$ and quadratic regression: $y = (275 \pm 0.5) \times 10^{-6} * x - (86.8 \pm 0.9) * 10^{-10} * x^2$. Key parameters are as follows: gradient strength = 10, cluster size = 5, protrusion amplitude = 1, protrusion period = 15.*

accompanied by its neighbor on account of strong cohesion, so the leading cell can only move backward a small distance, while it can move forward as far as its governing Gaussian allows. By contrast, the trailing cell travels nearly by itself, is seldom constrained by its neighbor, and so moves in either direction with nearly equal likelihood.

The mean cluster migration shown in Fig 6A for stochastic motion of the outermost cells contrasts with the stationary center of mass seen in Fig 3A for regular motion of those cells. This contrast can be evaluated, as shown in Fig 6B, where we plot longer simulations with and without stochastic extension, and with and without a cohesion gradient. Starting from the bottom of the plot, we plot in gray the earlier result that regular extensions, with any strength gradient, produce no motion of the cluster center of mass – i.e., the mean cluster speed is 0.

If the outermost cells wander stochastically without an internal cohesion gradient, the cluster will simply diffuse, equally likely in either direction, with a mean cluster speed of $(-0.1 \pm 0.1) \cdot 10^{-3}$ radii per timestep ($mr/\tau$ for brevity), as shown in blue in Fig 6B. When we include a cohesion gradient along with stochastic motion, however, we consistently see significant speeds toward the more cohesive end of the cluster as shown in red in Fig 6B, where the mean cluster speed is $(2.0 \pm 0.1)$ $mr/\tau$. Fig 6C includes a Mean Squared Displacement (MSD) analysis on 100 simulations. Migrating clusters without an internal cohesion gradient exhibit a linear trend indicating a diffusive or random migration pattern ($R^2 = 0.997$), while clusters with an internal gradient exhibit a quadratic trend indicating directed motion ($R^2 = 0.999$) [57]. These results taken together indicate that a cohesion strength gradient, when combined with intrinsic cellular stochasticity (described by the protrusion amplitude and period), generate a mechanism for directional migration. We further explore the impact of these key parameters on mean cluster speed in the next section.

## Quantitative testing of key parameters: gradient strength, cluster size, protrusion amplitude, and protrusion period

Equation (6) provides predictions for the persistent motion of a cluster in the limit of very strong leading cell cohesion and very weak trailing cell cohesion. We test these predictions by varying the equation's three available parameters: distance, $\Delta$, between leading neighbors (related to the cohesion gradient strength), number of cells, $N$, in a cluster the mean extension distance (proportional to the Gaussian standard deviation, $\sigma$), and the period, $T$, of extending protrusions.

In Fig 7, we display results of simulations varying each parameter, where each data point is an average over 100 independent trials. To assess whether results are dependent on details of cellular rheology, we display plots for the Standard Linear Solid (SLS) cellular model as well as the simpler Kelvin Voigt (KV) model. All simulations are computationally damped to mimic a low Reynolds number in a viscous cellular environment. To achieve this computational damping, we consider the Stokes limit for KV simulations (i.e., we reduce all velocities by 100% after every timestep). SLS simulations include internal viscous creep which creates an additional damping, so we reduce velocities less (by 25%) every timestep for those simulations. This produces exponential decay, as occurs in viscous damping. We find consistent qualitative behaviors provided that damping is strong; for weak damping, internal model springs (Fig 3) tend to oscillate.

Fig 7A confirms that the mean migration speed of a cluster grows with gradient strength. As described previously, all cells experience cohesive tension with their nearest neighbors, where the tension grows linearly from the trailing to the leading cell. In panel (a), we plot the speed of a 5 cell cluster as a function of the ratio between the highest and the lowest cohesive strength, which shows that cluster speed grows from zero, without a cluster gradient, to an

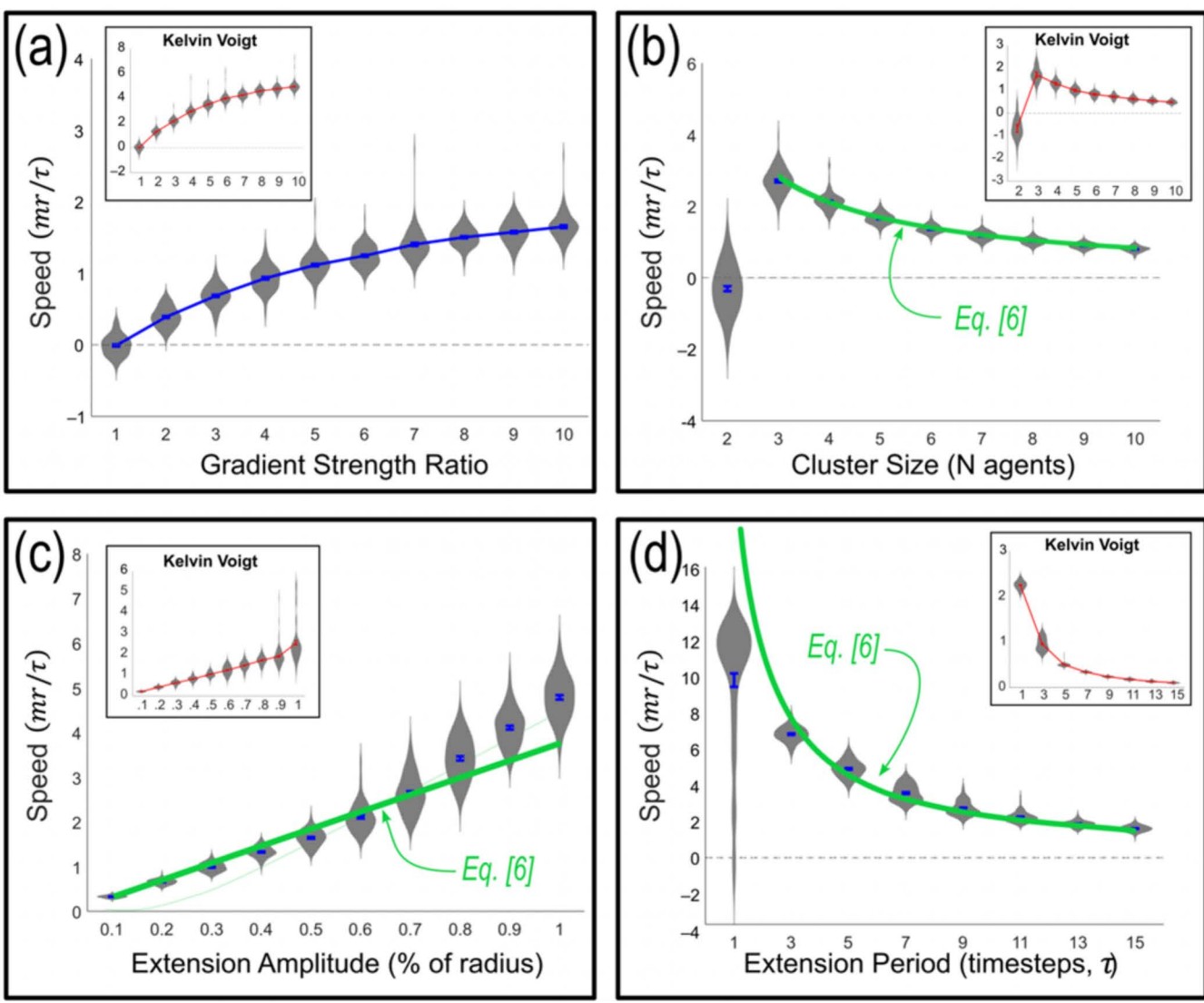

**Fig 7. Parametric response.** *Cluster speeds calculated from 100 simulations per parameter generated from stochasticity displacements of end agents only. Main plots show Standard Linear Solid (SLS), insets show Kelvin-Voigt (KV) model results. Violin plots show the distribution of all data point, and standard errors are shown for each parameter value. (a) Speed (unit distance relative to the radius of agents per computational timestep) vs cohesive gradient strength (the ratio of the strongest to weakest attraction in a linear gradient). Cluster speed grows from zero for gradient strength of 1 (equal leading and trailing cell cohesion), to an asymptote at large ratio. Cluster size = 5, protrusion amplitude = 0.5, protrusion period = 15. (b) Speed vs number of cells, N, in cluster produces a 1/N dependence as predicted by Eq. (6). Note N=2 has no gradient and migrates randomly. Gradient strength = 10, protrusion amplitude = 0.5, protrusion period = 15. (c) Speed vs protrusion amplitude (proportional to σ in Eq. (6)). Gradient strength = 10, cluster size = 5, protrusion period = 15. (d) Speed vs protrusion period of protrusion. Gradient strength = 10, cluster size = 5, protrusion amplitude = 0.5.*

asymptote at large gradient strength ratio. Qualitative behavior is unchanged with and without internal cellular creep (SLS vs. KV inset). Quantitatively, speeds are expressed in computational units of thousands of a cell radius per computation timestep (details appear in Models & Methods, and full code is included in S1 MATLAB Code 1 and S2 MATLAB Code 2). Since the SLS model contains an additional timescale associated with cellular creep, speed values differ compared with KV calculations.

In Fig 7B, we show the dependence on cluster size, *N*. Evidently, data from both models are well fit by the $1/N$ prediction from Eq. (6) using their respective parameters. This confirms

that increasing the number of cells increases the cluster inertia but does not affect the mechanism of driving. Put simply, the entire cluster is driven by a single source that is unaffected by increasing the number of cells: the bulk of cells in a cluster do not contribute to net motility in this model.

Fig 7C shows that the cluster speed grows monotonically as predicted by Eq. (6), but data show greater variation in speed with greater extension length, as exhibited both by longer violin plots at larger extension amplitudes and by standard error bars. We argue that in higher noise environments, more collision events would naturally occur between the outermost cells and their neighbors, and these collisions can either help the cluster move or hinder it depending on the direction of the collision. One of the assumptions made to derive Eq. (6) is that cells cannot overlap with another cell, and even with higher noise we maintain that rule in the simulations. Since the extension amplitude is governed by noise – i.e., by Gaussian standard deviation, $\sigma$, and consequently the fit to Eq. (6) worsens at larger extension amplitudes.

Finally, Fig 7D confirms the converse of speed growth with extension amplitude seen in panel (c). That is, by the same token that that cluster speed grows nearly proportionally to extension amplitude, we expect the inverse of cluster speed to grow with the inverse of extension speed – or that speed should grow like the inverse of extension period. Panel (d) supports this expectation, and displays improved fit for less frequent extensions, in keeping with the prior observation that more noise (here more frequent noise) in extensions produces more noise in cluster motion.

Apparently, the simple model depicted in Fig 5 that leads to Eq. (6) agrees with simulations of cluster migration, and in all cases, the KV model shown in insets agrees with the behaviors produced using the more realistic SLS model.

## Cluster speed is *reduced* by migration of interior cells

The essential proposition leading to Eq. (6) and supported by Fig 7, is that persistent motion of clusters of cells can be driven by a randomly wandering leader cell that is free on its leading edge and close to neighbors on its trailing edge. This differs from other collective migration phenomena, for example of bird flocks [58], fish schools [59], or bacteria swarms [60], in which all members of a collection actively participate [61].

We highlight this difference by allowing interior agents in a cluster to migrate, rather than only the outermost agents as in Fig 7. In Fig 8, we compare cluster speeds for three scenarios. These all use the more detailed SLS cellular model; similar results are found using the simpler Kelvin-Voigt Model.

First, in green, show the reference growth in speed with extension amplitude reproduced from Fig 7C. We compare that in red with speed vs. amplitude if all cells migrate in the presence of a cohesion gradient: this is exactly as in Fig 7C but allowing all cells to migrate according to the same Gaussian distribution. Evidently, cluster motion is prevented in this case. Finally, in blue we show that net cluster migration is rescued by allowing the endmost cells migrate as before but reducing Gaussian standard deviation of the central cells here by 75%.

Fig 8 demonstrates the surprising finding that the mechanism of cluster migration that we have explored here depends on suppressing internal cellular motion. There appears to be physiological evidence that leader-follower dynamics can effectively suppress cellular migration within a cluster [62,63]; for example adhesion molecules inhibit protrusion formation [64,65]. Whether this is an evolved response to promote cluster migration seems questionable. What we can conclude based on analytic and computational work is that randomly wandering leader cells in the presence of a cellular density gradient produce cluster migration, and that this migration is inhibited by participation of following cells. Ultimately, in all simulations

shown, we found that deterministic forcing (as shown in Fig 3) never produces collective migration, under any conditions, while stochastic forcing (Fig 8) can.

## Conclusion

Collective migration is a widely observed phenomenon across many cellular processes including morphogenesis, wound healing, cancer invasion, and tissue regeneration. Many different collective migration mechanisms have been proposed over the years including cell-cell and cell-substrate adhesion [50,66], leader-follower dynamics [67,68], contact inhibition of locomotion [69,70], and gradient sensing [71–73]. Many of these mechanisms work in tandem to generate effective collective motion and how they interact with one another is incompletely understood. Here we focus on the effects of cell-cell cohesion interactions produced by an external chemical gradient. We have performed a detailed kinematic analysis of cells in a cluster, and we have reached the following surprising findings. These differ from results reported for other collective phenomena.

1) Internal interactions within a cluster can contribute to net, external, motion of the cluster.

2) This effect requires individual cells to migrate stochastically: periodic displacements of cells do not produce net cluster motion.

3) The effect can arise when the leading cells are more closely spaced than the characteristic distance of stochastic migration. This spacing can be produced by an internal cohesion gradient.

4) The effect requires that migration of cells within a cluster be suppressed compared to migration of leading cells.

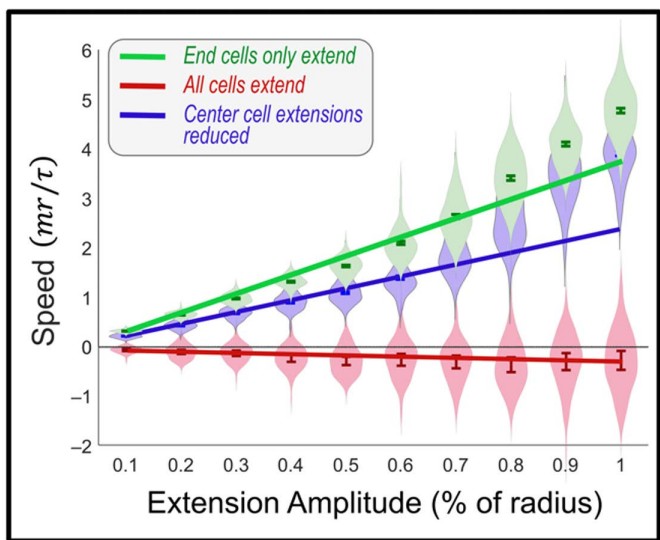

**Fig 8.** *Effect of internal migration on cluster motion.* *100 simulation means of cluster migration speed for three cases using Standard Linear Solid (SLS) model. Green: persistent motion of cluster is seen when only outermost cells in cluster wander stochastically. All other parameters from* Fig 7C. *Red: identical simulation in which all cells wander randomly, demonstrating that wandering of internal cells interferes with cluster migration. Blue: hybrid simulation in which outermost cells wander as in the green case, but all other cells migrate with σ reduced by 75%. Evidently, persistent cluster migration depends on suppression of wandering of interior cells.*

Our model has demonstrated a mechanism which explains why neural progenitors migrate along a growth factor gradient more effectively in clusters as opposed to single cells. This observation challenged the traditional understanding of chemotactic mechanisms in which individual cells sense their environment and growth factor binding triggers a signaling process guiding migration [74]. We have demonstrated that a gradient in *internal* connection strengths within a cluster of cells is sufficient to produce directional migration. This mechanism shares features with cohesotaxis, in which cell clusters migrate due to increased *external* traction stress produced by leading cells [75–77]. Fundamentally, cohesotaxis relies on an external stress gradient in the presence of symmetric cohesion between cells experiencing monopolar protrusions, while our analysis indicates that *internal* asymmetrical cohesion of cells experiencing nonpolarized protrusions in the presence of environmental noise can drive collective migration. Analogously, cohesotaxis is akin to a locomotive pulling a train, while this new mechanism is closer to metaboly, in which organisms propel itself following internal deformation.

This novel understanding of collective migration is essential to progressing research in cell replacement therapy. It has been well established that cell-cell adhesion molecules can play a critical role in morphogenesis and migration in tissue regeneration. Previous studies have shown an upregulation of cadherins by external growth factors as well as an increase in the directionality of cellular migration [17,19]. Our findings indicate that an internal cohesion gradient, stemming from differential external factors, leads to collective migration not previously described. We propose that improved understanding of this new mechanism may support future replacement cell transplantation therapies.

## Models and methods

We model our proposed differential cohesion mechanism using both the Kelvin-Voigt model and the more complex Standard Linear Solid model. Our models rely on three essential components 1) differential cohesion, 2) cellular stochasticity, and 3) environmental viscosity. A cohesion strength gradient provides the initial bias toward the positive direction by generating a stronger force at one end of the cluster. The stochastic forces are required to generate the initial motion in cells and prevent the system from reaching an equilibrium state. Environmental viscosity provides viscous drag forces which are directly proportional the cellular velocity, and since the stronger cohesion strength leads to faster velocities, the resulting faster drag force helps drive the cluster in the direction of the gradient. Details follow.

### Agent initialization

Simulations were performed using MATLAB software and object-oriented programming methods. N number of spherical agents were developed to represent individual cells within a cluster and track changes in key parameters over time. Constant parameters included mass ($m$), spherical radius ($r$), repulsive strength ($K_r$), and zone distance ($zonemax$) which defines the detection distance in which an agent will interact with any neighbors. Each agent is assigned an initial position ($x, y$) such that they are linearly conFigd with edges just touching their nearest neighbors and an initial velocity $(\dot{x})$ and/or acceleration $(\ddot{x}) = 0$. Based on the initial positions, linearly increasing cohesion strengths $(K_a)$ are assigned from user defined minimum and maximums, the ratio of which defines the gradient strength. In addition, an external parameter, $\eta_{environment}$ is defined as a value between 0 and 1 to represent the viscosity of the systems environment.

## Intercellular interactions

Intercellular interactions between agents and their nearest neighbors are governed by either the Kelvin-Voigt model:

$$\ddot{x}_n = \left[ \left( K_a x_{n-1} - \left( K_a + K_r \right) x_n + K_r x_{n+1} \right) - \eta \dot{x}_n \right] \tag{7}$$

or the Standard Linear Solid model:

$$\ddot{x}_n = \frac{1}{\tau} \left( K_a x_{n-1} - \left( K_a + K_r \right) x_n + K_r x_{n+1} \right) + \eta \dot{x}_n - \frac{1}{\tau} \ddot{x}_n \tag{8}$$

of viscoelasticity. In both cases, the force on a particular agent is determined by the relative positions of the neighbors and whether the initial agent experiences tension ($K_a$) or compression ($K_r$), therefore the K terms can be switched (as seen in the following equations).

$$\ddot{x}_n = \left[ \left( K_r x_{n-1} - \left( K_r + K_a \right) x_n + K_a x_{n+1} \right) - \eta \dot{x}_n \right] \tag{9}$$

$$\ddot{x}_n = \frac{1}{\tau} \left( K_r x_{n-1} - \left( K_r + K_a \right) x_n + K_a x_{n+1} \right) + \eta \dot{x}_n - \frac{1}{\tau} \ddot{x}_n \tag{10}$$

The new position of each agent $(t = 1)$ is dependent on the force, position, velocity, and acceleration at $(t = 0)$. The ODEs are solved using 4th order Runge Kutte integration, for the duration of the "relaxation" time (the time period between stochastic events), where the spring damper systems move toward a no stress state.

## Implementation of stochasticity

Our model contains stochasticity as a means of driving initial migration as the system begins in a desired resting state with agents just touching the nearest neighbors. There are two sources of stochasticity in our model: 1) the protrusion amplitude, which defines the distance an agent can move in a single time step, and 2) the protrusion period, defining the frequency of stochastic jumps within the simulation.

The protrusion amplitude describes the length of the lamellipodia or filopodia protrusion from the cell's edge. Mathematically, this parameter is defined as the standard deviation of the gaussian curve that determines both the direction of the protrusion and the amplitude. The mean for the curve is always set to zero to maintain equal opportunity for protrusion formation in either direction. In our simulations, when stochastic protrusions occur, a random value is generated from the gaussian curve. If the value is within the bounds set by the neighbors' distances, then it is accepted, and the agent is moved to the new location immediately. If the value is considered out of bounds, or would cause agents to overlap, then the value is rejected, and no stochastic movement occurs.

The protrusion period describes the frequency of protrusion formation, as well as the corresponding relaxation time. For example, if a simulation was run for 2000 timesteps with a protrusion period of 15, every 15 steps a protrusion is randomly generated and either accepted or rejected based on the model's constraints. Following the protrusion period is a relaxation time until the next protrusion period. During this time, agents can respond to the sudden stress generated by the separation of agents and attempt to return to their desired resting state of just touching. Biologically, this time describes the translocation and retraction times in individual cell motility. Higher protrusion periods correspond to less frequent stochastic movement and longer relaxation times.

## Sources of viscosity

We have two sources of viscosity in our model: 1) the intracellular viscosity as described by the damper component of the viscoelastic equations, and 2) the environmental viscosity. The environmental viscosity is an additional user defined parameter in our model and for computational simplicity, it is a value between 0 and 1 that is multiplied by the initial velocity (and acceleration) prior to the numerical calculations. A value of 0 describes a purely viscous fluid, while a value 1 represents no viscosity in the fluid and no damping on cell motion. All the data shown previously has been generated under the assumption that the environmental fluid is very viscous, or with an environmental viscosity of 0.

## Statistical analysis

All simulation parameters were run in replicates of 100 times to account for stochastic variations. A Mean Squared Displacement (MSD) analysis was conducted on comparisons between collective migration with and without a gradient to determine persistent directional migration. Multiple Student t-test were performed on Kelvin-Voigt and Standard Linear Solid data. Our analysis revealed no significant differences between groups ($p > 0.05$), indicating that both models are suitable for describing the proposed mechanism.

## Supplementary information:

**S1 Matlab Code 1. Main_Simulation.m is the script that contains all the steps necessary to generate the data shown in Figs 3, 6–8** . The code is written using object orientated programming and is required to work in conjunction with S2 Matlab Code 2: Agent.m.
(DOCX)

**S2 Matlab Code 2. Agent.m is the script where the object, representing an individual cell, and all the corresponding methods are developed** . This code is required to run S1 Matlab Code 1: Main_Simulation.m.
(DOCX)

## Author contributions

**Conceptualization:** Larissa M. Oprysk, Maribel Vazquez, Troy Shinbrot.

**Formal analysis:** Larissa M. Oprysk.

**Funding acquisition:** Maribel Vazquez.

**Investigation:** Larissa M. Oprysk.

**Software:** Larissa M. Oprysk.

**Supervision:** Maribel Vazquez, Troy Shinbrot.

**Visualization:** Larissa M. Oprysk, Troy Shinbrot.

**Writing – original draft:** Larissa M. Oprysk.

**Writing – review & editing:** Larissa M. Oprysk, Maribel Vazquez, Troy Shinbrot.

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
