## [Decision Letter · Decision Letter 0]

17 Jul 2024

Dear Dr. Vazquez,

Thank you very much for submitting your manuscript "Internal Cohesion Gradient as a Novel Mechanism of Collective Cell Migration" for consideration at PLOS Computational Biology.

As with all papers reviewed by the journal, your manuscript was reviewed by members of the editorial board and by several independent reviewers. In light of the reviews (below this email), we would like to invite the resubmission of a significantly-revised version that takes into account the reviewers' comments.

We cannot make any decision about publication until we have seen the revised manuscript and your response to the reviewers' comments. Your revised manuscript is also likely to be sent to reviewers for further evaluation.

Sincerely,

Philip K Maini

Academic Editor

PLOS Computational Biology

Jason Haugh

Section Editor

PLOS Computational Biology

Reviewer's Responses to Questions

**Comments to the Authors:**

Reviewer #1: Please see the attached PDF.

Reviewer #2: The authors claim to present a novel mechanism of directed collective cell migration. Cells actively and stochastically migrate but are also connected by forces, modelled by springs. The spring constants are graded across the 1D set of cells. Cells at the stiffer end become closer together and it is assumed that protrusions can only reach as far as the nearest neighbours. This means that the cell at the stiffest end has a bias for protrusions away from the group of cells, whereas the cell at the less stiff end has negligible bias in protrusions, since it has room to extend in both directions. Thus, the protrusions on the leading cell generate a net force on the group of cells in the direction out from the leading cell. This leads to a net migration towards the stiffer end. The authors assert that the mechanism may be of relevance in retinal regeneration in which there is a FGF gradient which may affect cell stiffness.

I have some comments:

1. The authors should explain how their mechanism differs from cohesotaxis (see references below) and cite the relevant literature on that.

Example cohesotaxis references:

Weber, G.F., Bjerke, M.A. and DeSimone, D.W., 2012. A mechanoresponsive cadherin-keratin complex directs polarized protrusive behavior and collective cell migration. Developmental cell, 22(1), pp.104-115.

Zhao, J., Cao, Y., DiPietro, L.A. and Liang, J., 2017. Dynamic cellular finite-element method for modelling large-scale cell migration and proliferation under the control of mechanical and biochemical cues: a study of re-epithelialization. Journal of The Royal Society Interface, 14(129), p.20160959.

2. The authors suggest a gradient in the spring stiffness throughout the tissue, but is that actually necessary? Would it be sufficient just to have the leading cell attached by a stiffer connection?

3. Line 34 ingrate -> integrate

4. Lines 104 & 105- rewrite; it appears the final term is described twice and the descriptions differ.

5. Line 128- replace semicolon with comma

6. Paragraph (lines 148-156)- There should be a better description of exactly what is done here. For example, what are the minimum and maximum cohesion strengths? What does cohesion strength mean (K_a?)? This is mentioned later, so if you do not want to define it here, then refer to the relevant section later in the paper. What does ‘relax’ mean? Do you integrate the equation on line 412/ line 414, and for how long? Precisely what parameter values are used.

7. Line 153- remove ‘least’

8. Legend figure 3: define protrusion amplitude, period and gradient amplitude.

9. Line 197 leading cell -> leading cells

10. Line 198- you say the lowest cell, but it looks like the highest cell. Should Fig. 5 a) and b) be flipped on a horizontal axis to agree with the descriptions… (Also line 202/203 upwards and downwards seem to differ from figure.)

11. Line 272 Fig. 5(b) -> Fig. 6(b)

12. Paragraph (lines 288-296). If you reduce the velocities to zero after every timestep, then do the results not depend on your timestep? If not, please explain why? If so, this should be acknowledged- it seems unphysical.

13. Line 372- The authors should put the work in context by discussing of other means of collective migration, especially those, such as cohesotaxis, which may be quite related.

14. Line 385- the authors should add a reference to the end of the first sentence on this page.

15. Line 412- Surely this depends on whether the spring to n-1 or n+1 is under tension (i.e. Ka and Kr may be switched).

16. Line 444 multiplied to -> multiplied by

Reviewer #3: This manuscript describes an interesting model that demonstrates how collective cell migration could result from asymmetry in internal cohesion between adjacent cells in a cluster of cells. The authors make a solid argument that modeling and studying the biophysical mechanisms of collective cell migration is critical to understanding a wide range of physiological and pathological tissue adaptations, from embryonic morphogenesis to tumorigenesis. The authors also persuasively argue that their model takes an important first step towards the design of better injectable stem cell therapies, wherein donor cell migration and integration with native tissue can be optimized according to the insights provided by their model. The model is well described, and the results are clearly presented. However, as enumerated below, I have the following concerns/confusion about some of the terminology used, the description of the model assumptions, and interpretation of the model predictions.

1. Abstract: The authors introduce the important concept of “asymmetry near the leading cell” but it is not clear at this point what asymmetry they are referring to. The authors should clarify, “..asymmetry [of what?]…” in this key sentence.

2. Introduction: A related point to the one above, is that the authors should relate the concept of “internal cohesion gradient” to the biology of cell-cell adhesion. Could the internal cohesion gradient, which is a lynchpin of the model, represent gradients of cadherin receptors in neighboring cells? Or gradients of activated cadherin receptors across neighboring cells? If this is the case, the authors should try to cite literature that reports such biological observations because that will help demonstrate the biological relevance of their model.

3. Introduction: (Page 2, last paragraph): The sentence that begins, “In our lab we study…” seems out of place in a manuscript and reads more like a grant. My suggestion is to begin this sentence with: “We have previously shown in vitro that in the presence…”

4. Introduction: The reproduction of Figure 1 from a different published paper may not be allowed by the journal and I think it would be better to just cite the paper where this finding was published.

5. Introduction: (Page 3, first paragraph): The sentence that begins: “These factors do not appear to affect persistent cell migration of individual cells,…” is confusing. What do “these factors” refer to in this sentence? Cadherins or growth factors?

6. Introduction: (Page 3, last paragraph): The authors state that “…cells can generate an external force that individual cells do seem to not feel.” This is confusing. How do we know that cells don’t “feel” an external force? Maybe it would help to define what is meant by “external force”.

7. Introduction: (Page 4, last paragraph): This is the first place where the authors introduce some key assumptions of their model, and I think they should expand on the discussion of the model assumptions and limitations throughout the Results, as well. One major assumption stated here is that substrate stresses are assumed to be identical along the length of the cluster. The authors should justify this assumption and/or explain whether this is a limitation of their model, given what has been experimentally observed in biology. Also, what does “omnidirectional wandering” mean?

8. Results: (Page 7, first paragraph): This sentence is confusing: “…however if either mechanism is activated periodically in time using alternating for-ward and rear-ward distances periodically in time, no collective motions of clusters is seen…” What does “periodically in time” refer to? What does “alternating for-ward and rear-ward distances” refer to? Are these distances set lengths? What is the period of time during which the alternating is occurring? Also, what does “migrate stochastically” mean? Does an individual cell randomly decide to move forward and backward according to some period of time that is set in the model, or is that period of time stochastic, as well?

9. Results: Most of the results sub-heading titles are confusing and do not adequately describe the results that are being presented in that sub-section. For example, what does “Baseline” refer to in the 2nd sub-heading title? Similarly, what does “Analysis: baseline case plus diffusion of out cells” mean (specifically, what is being analyzed?). Also, “Quantification” as a sub-heading is confusing. (Quantification of what? And were the prior results not also “quantification”?). And, “Qualitative confirmation” (of what?). And, “Quantitative parameteric test” (of what?). Revising the Results sub-section headings will help improve clarity and help the reader better understand how each section builds (or relates to) the prior results.

10. Results: (Page 7, last paragraph): The sentence: “As expected, the cells with the strongest (weakest) tensile moduli deviate least (most) from the least center of mass…” is confusing.

11. Results (Figure 7 and Page 15, last paragraph): The authors inconsistently refer to “Gradient strength ratio” and “Gradient strength” in the figure caption and in the text. Do these terms mean the same thing? Also, it should be clear to the reader that the authors are referring to the “cohesive gradient strength” or “cohesive gradient strength ratio”, and these terms should be clearly defined in the manuscript.

12. Results: Why did the authors initially chose to represent 5 cells in their model?

13. Results: (Page 18, first paragraph): The authors state that the model makes the surprising prediction that the mechanism of cluster migration that they have explored depends on “suppressing internal cellular motion”. This is based on the fact that when they reduce the Gaussian standard deviation of the central cells by 75% net cluster migration is rescued. What happens when they reduce it by even more (like 95%) or by less (like 1%)? This is an important prediction and showing a full sensitivity analysis would help the reader understand how significant this effect is on cluster migration speed.

14. Conclusion: Revisiting my comment #2 above, I think the impact of the paper would be elevated by discussing whether cadherin expression (and/or activation) gradients across neighboring cells have been examined and how that data relates to the “cohesive gradient strength” which is the central prediction of this model.

**Have the authors made all data and (if applicable) computational code underlying the findings in their manuscript fully available?**

Reviewer #1: Yes

Reviewer #2: Yes

Reviewer #3: Yes

PLOS authors have the option to publish the peer review history of their article (what does this mean? ). If published, this will include your full peer review and any attached files.

**Do you want your identity to be public for this peer review?** For information about this choice, including consent withdrawal, please see our Privacy Policy .

Reviewer #1: No

Reviewer #2: No

Reviewer #3: No
---

## [Decision Letter · Decision Letter 1]

22 Nov 2024

PCOMPBIOL-D-24-00775R1Internal Cohesion Gradient as a Novel Mechanism of Collective Cell MigrationPLOS Computational Biology  Dear Dr. Vazquez, Thank you for submitting your manuscript to PLOS Computational Biology. After careful consideration, we feel that it has merit but does not fully meet PLOS Computational Biology's publication criteria as it currently stands. Therefore, we invite you to submit a revised version of the manuscript that addresses the points raised during the review process. Please submit your revised manuscript within 30 days Jan 22 2025 11:59PM. If you will need more time than this to complete your revisions, please reply to this message or contact the journal office at ploscompbiol@plos.org. Please include the following items when submitting your revised manuscript: * A rebuttal letter that responds to each point raised by the editor and reviewer(s). You should upload this letter as a separate file labeled 'Response to Reviewers'. This file does not need to include responses to formatting updates and technical items listed in the 'Journal Requirements' section below. * A marked-up copy of your manuscript that highlights changes made to the original version. You should upload this as a separate file labeled 'Revised Manuscript with Track Changes'. * An unmarked version of your revised paper without tracked changes. You should upload this as a separate file labeled 'Manuscript'. If you would like to make changes to your financial disclosure, competing interests statement, or data availability statement, please make these updates within the submission form at the time of resubmission. Guidelines for resubmitting your figure files are available below the reviewer comments at the end of this letter.

We look forward to receiving your revised manuscript.

Kind regards,

Philip K Maini

Academic Editor

PLOS Computational Biology

Jason HaughSection EditorPLOS Computational Biology 

Feilim Mac Gabhann

Editor-in-Chief

PLOS Computational Biology

Jason Papin

Editor-in-Chief

PLOS Computational Biology

**Reviewers' comments:**

Reviewer's Responses to Questions

**Comments to the Authors:**

Reviewer #1: Please see the attached PDF.

Reviewer #2: The authors have successfully addressed my comments and I believe the paper is now acceptable for publication.

**Have the authors made all data and (if applicable) computational code underlying the findings in their manuscript fully available?**

Reviewer #1: **No: ** Relevant MATLAB codes have been provided, but the raw data that have produced the figures have not been sent and there is no information on how to obtain them. The authors should upload the raw data to a repository (e.g., Dryad) and cite a link to the dataset in the manuscript.

Reviewer #2: Yes

PLOS authors have the option to publish the peer review history of their article (what does this mean? ). If published, this will include your full peer review and any attached files.

**Do you want your identity to be public for this peer review?** For information about this choice, including consent withdrawal, please see our Privacy Policy .

Reviewer #1: No

Reviewer #2: No

**Figure resubmission:**
---

## [Editor Report · Decision Letter 2]

7 Jan 2025

Dear Dr. Vazquez,

We are pleased to inform you that your manuscript 'Internal Cohesion Gradient as a Novel Mechanism of Collective Cell Migration' has been provisionally accepted for publication in PLOS Computational Biology.

Best regards,

Philip K Maini

Academic Editor

PLOS Computational Biology

Jason Haugh

Section Editor

PLOS Computational Biology

---

## [Editor Report · Acceptance letter]

PCOMPBIOL-D-24-00775R2

Internal Cohesion Gradient as a Novel Mechanism of Collective Cell Migration

Dear Dr Vazquez,

I am pleased to inform you that your manuscript has been formally accepted for publication in PLOS Computational Biology. Your manuscript is now with our production department and you will be notified of the publication date in due course.

With kind regards,

Zsofia Freund
